# Investigating Patients with Pulmonary Hypertension Under Detector-Based Spectral Computed Tomography

**DOI:** 10.3390/diagnostics15091069

**Published:** 2025-04-23

**Authors:** Hsien-Fu Cheng, Yu-Pin Chang, Jyh-Wen Chai

**Affiliations:** 1Department of Radiology, Taichung Veterans General Hospital, Taichung 407219, Taiwan; chenghsienfu@gmail.com (H.-F.C.); hubt@mail.vghtc.gov.tw (J.-W.C.); 2Premium Health Examination Center, Tungs’ Taichung MetroHarbor Hospital, Taichung 43503, Taiwan

**Keywords:** computed tomography, dual energy, pulmonary hypertension, spectral detector CT, iodine density, effective atomic number (Zeff), chronic thromboembolic PH (CTEPH), detector-based spectral computed tomography

## Abstract

**Background**: Pulmonary hypertension (PH) is characterized by elevated pressure in the pulmonary artery. Currently, most dual-energy CT (DECT) research focuses on the application of iodine mapping in pulmonary embolism. However, little attention is paid to the parametric mapping of the lung parenchyma of PH. **Methods**: In total, 156 cases undergoing thoracic DECT from 2021 August to 2023 February were surveyed. For each case, the iodine density (Iod) and effective atomic number (Zeff) of four different levels of the lung, along with the iodine density of the pulmonary artery and aorta, were measured. The measured parameters and their derivatives were compared between PH cases and normal controls and between chronic thromboembolic PH (CTEPH) and non-CTEPH cases. **Results**: Region of interest (ROI)-Zeff was statistically lower in the PH group as compared to the normal controls on each level. The ratio of PA-iod/ROI-iod was significantly higher in the PH group than in the normal controls. ROI-iod was statistically lower in the CTEPH cases as compared with the non-CTEPH cases on each level. The CTEPH cases demonstrated a higher PA-iod/ROI-iod value as compared with the non-CTEPH cases. **Conclusions**: The PA-iodine density and effective Z of spectrum CT could serve as valuable imaging parameters for the diagnosis and characterization of PH and CTEPH.

## 1. Introduction

Pulmonary hypertension (PH) is a severe and progressive condition characterized by high blood pressure in the pulmonary arteries, affecting approximately 1% of the global population [1]. It can lead to right heart failure, pulmonary edema, Eissenmenger syndrome (in the presence of a shunt), and, ultimately, death if left untreated [2]. According to the study conducted in Giessen, Germany, investigating survival rates among different pulmonary hypertension groups without receiving lung transplants, the overall average survival rates were about 86% at 1 year, 67% at 3 years, and 54% at 5 years [3].

As characterized according to the sixth World Symposium on Pulmonary Hypertension, an individual is diagnosed with pulmonary hypertension when the resting pulmonary mean arterial pressure surpasses 20 mmHg, as measured by right heart catheterization [4]. According to the ESC/ERS guidelines, PH is clinically classified based on hemodynamic characteristics and a comprehensive set of clinical investigations. The clinical classification defines five subgroups: pulmonary arterial hypertension (PAH, group 1), PH associated with left-sided heart disease (group 2), PH associated with lung diseases or hypoxia (group 3), chronic thromboembolic PH (CTEPH) and other pulmonary artery obstructions (group 4), and PH with unclear/multifactorial mechanisms (group 5). While all groups share an increase in the mean pulmonary arterial pressure (mPAP), the hemodynamic classification differs and could be categorized into precapillary PH, isolated postcapillary PH, combined precapillary and postcapillary PH, and exercise PH according to their PH groups, respectively [5].

Although right heart catheterization (RHC) is the gold standard for diagnosing and classifying PH, it is an invasive procedure and requires expertise and meticulous methodology following standardized protocols. It could be accompanied by peri- and post-interventional complications and does not provide morphologic information [6]. Current noninvasive diagnostic methods for pulmonary hypertension (PH) include echocardiography, magnetic resonance imaging (MRI), and computed tomography (CT). Each modality, however, has its limitations. Echocardiography, although widely used, often inaccurately estimates pulmonary arterial pressures compared to right heart catheterization (RHC) and struggles to visualize pulmonary arteries directly, which can lead to missed diagnoses of conditions like pulmonary artery stenosis or thromboembolic disease [7,8]. MRI, while providing detailed anatomical and functional data, is limited by its high cost, prolonged scan times, sensitivity to motion artifacts, and high requirement of patient cooperation [9,10,11]. CT, on the other hand, is primarily an anatomical imaging modality that offers limited functional information and exposes patients to ionizing radiation, raising concerns, especially for those requiring multiple scans [12,13,14]. CT also carries potential nephrotoxicity from contrast media use, particularly in patients with pre-existing kidney disease [15,16,17]. These limitations highlight the need for careful selection and interpretation of imaging modalities in the diagnosis and management of PH.

It is well-documented that pulmonary perfusion decreases in cases of chronic thromboembolic pulmonary hypertension (CTEPH) [18,19,20]. However, even in non-CTEPH PH, studies have shown significant decreased pulmonary perfusion compared to normal controls [21,22,23]. Additionally, lung perfusion scintigraphy in idiopathic PAH patients revealed a distinct patchy pattern, further highlighting significant differences in perfusion patterns between PH patients and the normal population [24].

Dual-energy computed tomography (DECT) is an advanced imaging technique that uses two distinct X-ray energy levels to capture detailed images, enabling the differentiation of materials based on their distinct attenuation properties at different energies [25]. This technology allows improved image quality, reduced scanning time, and, most importantly, enhanced tissue characterization [26]. There are different types of dual-energy CT scanners, one of which is detector-based spectral computed tomography (SDCT). This type uses two layers of detectors to simultaneously capture low and high-energy image data [27].

The introduction of DECT has made it possible to map pulmonary perfusion by exploiting the different absorption characteristics of iodine and lung parenchyma [28]. Iodine density images are regarded as a reliable surrogate measurement for assessing organ perfusion [29]. They have demonstrated their capability to offer insights into pulmonary perfusion in both acute pulmonary embolism and chronic thromboembolic pulmonary hypertension (CTEPH) [30,31], showing accuracy comparable to or exceeding that of ventilation/perfusion scintigraphy (V/Q scintigraphy) [32].

Besides these advancements, we wonder if DECT parameters, such as iodine density and Zeff, play a role in distinguishing PH patients from the normal population. This study aimed to determine whether iodine density and Zeff derived from SDCT could differentiate PH patients—particularly CTEPH—from healthy individuals and from other PH subtypes.

## 2. Materials and Methods

### 2.1. Patient Population

A total of 156 cases with CT pulmonary angiogram (CTPA) conducted on spectrum CT in our institution from 2021 August to 2023 February were collected. A total of 99 patients were excluded due to severe infection/inflammation with marked opacification of lung parenchyma, lung cancer, lung metastatic lesions, or severe COPD (emphysema extent >50% total lung parenchyma via visual assessment).

Among the remaining patients, those who met the following two criteria were included in the pulmonary hypertension (PH) group: (1) underwent right heart catheterization during the same hospitalization as the CTPA examination and (2) had a mean pulmonary arterial pressure (mPAP) >25 mmHg at rest. Although the 6th World Symposium on Pulmonary Hypertension recommends a threshold of >20 mmHg for PH diagnosis, our study adopted the >25 mmHg cutoff based on our institutional practice for improved diagnostic specificity.

The control group consisted of cases with normal right heart echocardiography and normal right ventricular pressure. In total, 33 patients were enrolled as the PH group and 24 patients as the normal control group.

Of the 33 PH patients, 9 were clinically diagnosed with chronic thromboembolic pulmonary hypertension (CTEPH) based on clinical history and serial CT pulmonary angiography or right heart catheterization. The remaining 24 patients had PH from other causes and were categorized as the non-CTEPH group.

All included patients were newly diagnosed at the time of DECT examination. None of the non-CTEPH PH patients had received targeted pharmacological treatment before imaging. The CTEPH patients had received anticoagulant therapy for pulmonary embolism prior to imaging, but none had undergone pulmonary embolectomy or balloon pulmonary angioplasty (Figure 1).

### 2.2. Dual-Energy CT Pulmonary Angiography Imaging and Analysis

CT data were acquired on a clinically available detector-based spectral CT (IQon, Philips Healthcare, Best, The Netherlands). All the patients received an intravenous 100 mL bolus of contrast media (Omipaque, 350 mg Iodine/mL, GE Healthcare; Xenetix, 350 mg Iodine/mL, Guerbet) followed by a 60 mL normal saline injection at a flow rate of 3–4 mL/s. Scanning was initiated with a delay of 5 s after an attenuation of 150 HU was reached in the main pulmonary artery (MPA). Image reconstructions were performed under a Philips workstation (IntelliSpace Portal 11, Philips Healthcare). The images were reconstructed in axial orientation with a slice thickness of 1 mm and a slice overlap of 0.5 mm. In addition to conventional images, iodine density maps and effective atomic number (Zeff) images were also reconstructed under the postprocessing software workstation.

All quantitative measurements were performed on the axial images on the workstation. Among all 57 cases, each measurement, either iodine density or effective Z, was performed with an 80 to 100 mm^2^ circular area region of interest (ROI). The measurement of iodine density (ROI-iod) and effective Z(ROI-Zeff) of the target lung parenchyma was performed at 24 standardized locations, which were the anterior, mid, and posterior aspects of the bilateral lungs at four predefined levels (A: aortic arch, B: carina, C: right pulmonary artery, D: 2 cm above the diaphragm) (Figure 2). An additional E (either ROI-iod or ROI-Zeff) was calculated from the total average of 24 ROI-iod and ROI-Zeff of one subject. Furthermore, iodine densities of the pulmonary trunk (PA-iod) and the ascending aorta (Aa-iod) were also measured at the identical level. Notably, major vessels were intentionally excluded from these measurements. If any area of interest was obscured by suspected artifacts or dense pleuroparenchymal disease, such as consolidation or pleural effusion, those ROI measurements were excluded from the analysis. Ratios of PA-iod/ROI-iod and Aa-iod/ROI-iod were further calculated [28].

A single experienced radiologist, specialized in pulmonary and DECT imaging for 5 years, conducted all the measurements subsequent to a training session involving 20 cases that were not part of this study. Importantly, the radiologist remained unaware of both the patients’ diagnoses and their pulmonary pressure condition during the spectral assessment.

This study was approved by the institutional review board at the Taichung Veterans General Hospital, Taichung, Taiwan (CE24287B).

### 2.3. Statistical Analysis

We aimed to figure out if there are significant differences in ROI-iod, ROI-Zeff, PA-iod/ROI-iod, and Aa-iod/ROI-iod between the PH and normal control groups, as well as between the CTEPH and non-CTEPH groups. The Mann–Whitney U test was utilized. The data are presented as median (IQR), and all analyses were conducted using the SPSS 22 statistical software program by using the SPSS 22 statistical software program (SPSS, Inc., Chicago, IL, USA). A *p*-value of less than 0.05 was deemed significant.

## 3. Results

### 3.1. Study Population

The PH group comprised 33 patients [10 men and 23 women; mean age of 62 (44–80.5 years old)]; whereas the healthy group consisted of 24 subjects (12 men and 12 women) with a mean age of 60 years old (39.25–71.5 years old) [Table 1].

Among the 33 patients with pulmonary hypertension, 9 were classified as group 4 PH (chronic thromboembolic pulmonary hypertension, CTEPH), and the remaining 24 were classified as other subgroups (Table 2).

Measurement of pulmonary ROI-iod was conducted at 1332/1368 (97.4%) standardized locations, with 36 (2.6%) ROI-iods (30 in PH cases and 6 in normal controls) excluded due to plueroparenchymal disease or artifacts.

### 3.2. Differentiation Between PH Patients and Controls

A comparison of the spectral parameters between PH patients and normal controls is summarized in Table 3 and Figure 3. ROI-Zeff was statistically lower in the PH group at each level, as well as the total lung average (ROI-Zeff E, 9.80 vs. 10.28, *p* < 0.001). On the other hand, the ratio of PA-iod/ROI-iod was significantly higher in the PH group than in the normal controls, either at each selected level or the total lung average. The mean PA-iod/ROI-iod at the total lung average of the PH group and normal controls was 14.70 vs. 12.34 (*p* = 0.005).

There was no statistically significant difference in the PH group compared with the controls in ROI-iod and AO-iod/ROI-iod, either at each selected level or the total lung average.

### 3.3. Differentiation Between CTEPH Patients and Non-CTEPH Patients

The results of the comparison between the CTEPH group and the non-CTEPH group are summarized in Table 4 and Figure 4. ROI-iod was statistically lower in the CTEPH cases as compared with the non-CTEPH cases at each level and the total lung average (each *p* < 0.05). In the evaluation of PA-iod/ROI-iod, the CTEPH cases demonstrated higher values as compared with the non-CTEPH cases at each selected level and the total lung average (all *p* < 0.005). There was no significant difference in ROI-Zeff between the CTEPH cases and the non-CTEPH cases (all *p* > 0.05), nor the Aa-iod/ROI-iod (all *p* > 0.05).

To further illustrate these findings graphically, Figure 5 displays coronal images comparing the iodine density and Zeff among the normal controls, PH patients, and CTEPH patients.

## 4. Discussion

Overview of Key Findings

This study represents a pioneering effort in evaluating the utility of SDCT for assessing pulmonary hypertension by analyzing iodine density and Zeff in lung parenchyma. Our primary findings are as follows:The ratio of iodine density in the main pulmonary artery to that in the lung parenchyma (PA-iod/ROI-iod) is significantly elevated in PH patients compared to healthy controls.Among PH subtypes, patients with chronic thromboembolic pulmonary hypertension (CTEPH) exhibit even higher PA-iod/ROI-iod ratios than those with other forms of PH.The effective atomic number (Zeff) is notably lower in PH patients relative to controls.

Interpretation of Iodine Density Findings

The PA-iod/ROI-iod ratio emerged as a key metric in our study for evaluating vascular perfusion abnormalities in PH. We observed significantly higher PA-iod/ROI-iod ratios in the PH patients compared to the normal controls (mean PA-iod/ROI-iod E: 14.70 vs. 12.34, *p* = 0.005), indicative of increased iodine uptake in the central pulmonary arteries and/or decreased uptake in the peripheral lung parenchyma.

These findings align with previous research by Seyed Ameli-Renani et al. [28], which can be attributed to increased vascular resistance and impaired vascular compliance in PH. The elevated pulmonary arterial pressure in PH leads to prolonged contrast transit time and pooling of contrast material in the central pulmonary arteries. Additionally, the redistribution of pulmonary blood flow due to small-vessel remodeling and endothelial dysfunction may contribute to decreased iodine uptake in the lung parenchyma.

It is also noteworthy that most previous studies on DECT-based lung perfusion were performed using dual-source CT systems, while detector-based spectral CT remains less frequently studied. Although both approaches can generate iodine material images, their interchangeability across different platforms has not been fully validated. Our results confirm that perfusion-related parameters like PA-iod/ROI-iod can be reliably assessed using SDCT, thus expanding the clinical applicability of dual-energy techniques for PH evaluation.

From a clinical perspective, the PA-iod/ROI-iod ratio may serve as a noninvasive imaging parameter for assessing hemodynamic changes in PH. Future studies could explore its correlation with invasive right heart catheterization parameters to determine its potential role in disease stratification and early diagnosis.

Distinction Between CTEPH and Non-CTEPH

Furthermore, among PH subtypes, patients with chronic thromboembolic pulmonary hypertension (CTEPH) exhibit even higher PA-iod/ROI-iod ratios compared to other forms of PH. This disparity is probably due to the significantly reduced contrast delivered to peripheral lung parenchyma because of thromboembolic material in pulmonary arteries in CTEPH, resulting in a higher iodine concentration in the central, larger-sized pulmonary arteries [33,34,35]. The distinct vascular remodeling and altered hemodynamics in CTEPH, characterized by persistent thromboembolic obstructions, subsequent pulmonary hypertension, and increased pulmonary vascular resistance, exacerbate this extraordinary contrast distribution in the pulmonary vasculature and lung parenchyma [5,34,35,36,37,38].

These findings support the utility of DECT-derived iodine density ratios in capturing underlying vascular pathology, enabling the differentiation of CTEPH from other PH subtypes and offering valuable insights for noninvasive work-up of PH with uncertain etiology.

Role of Effective Atomic Number (Zeff) Imaging

While iodine-based images provide a snapshot of perfusion, recent studies suggest that effective atomic number (Zeff) imaging may offer complementary value in detecting pulmonary parenchymal changes.

Zeff represents the average atomic number of a compound or mixture, considering the relative proportion of its constituent elements [39]. Materials with higher Zeff interact more strongly with radiation, absorbing more energy from X-rays, gamma rays, or other forms of electromagnetic radiation [40]. In clinical imaging, high-Zeff materials, like iodine-based contrast agents, are utilized to enhance contrast in soft tissues or highlight specific organs [41].

In the current DECT, iodine material images were generated by applying two-material (iodine–water) or three-material decomposition (iodine–soft tissue–air) of the acquired DECT data. Ke Li et al. [42] identified two major limitations of iodine-based images. First, materials with effective atomic numbers different from the basis materials can contribute to iodine images, potentially leading to inaccurate representations of iodine distribution. Second, iodine images do not reflect lung tissue mass, whereas pulmonary blood volume (PBV) is the volume of blood perfused per unit mass of lung tissue.

In our study, we observed that the effective atomic number (ROI-Zeff) was statistically lower in the PH group compared to the controls (Zeff E: 9.80 vs. 10.28, *p* < 0.001). This reduction in Zeff may be attributed to several underlying pathophysiological mechanisms. Given that Zeff reflects the average atomic number of tissue, a decrease suggests alterations in the pulmonary parenchymal composition. One possible explanation is the reduction in pulmonary microvascular perfusion due to vascular remodeling, leading to decreased blood volume in the lung parenchyma. Since blood contains higher Z elements (e.g., iron in hemoglobin and iodine in contrast), reduced perfusion could lower the overall Zeff of the lung parenchyma tissue. Additionally, interstitial thickening and subtle fibrosis reported in PH patients may contribute to the decrease in Zeff by increasing the proportion of low-Z elements, such as carbon, hydrogen, and oxygen.

Physiologic Interplay Between Iodine Density and Zeff

Although iodine is a high-Z element and does contribute to Zeff values, the effective atomic number is derived from the averaged atomic composition within each voxel, including not only contrast medium but also intrinsic tissue characteristics, such as air content, interstitial matrix, fibrosis, and blood. In CTEPH, although the iodine density was significantly lower compared to non-CTEPH due to segmental perfusion defects, the volume and extent of these defects may have been insufficient to significantly alter the overall lung Zeff.

Additionally, both CTEPH and non-CTEPH groups fall under the PH spectrum and are likely to share certain degrees of pulmonary vascular remodeling and parenchymal changes. However, the spatial distribution and severity of these alterations may differ. While the localized perfusion defects characteristic of CTEPH and the segmental nature of embolic obstruction may result in areas with lower Zeff, when averaged across the lung, these regional changes may be diluted.

On the other hand, in non-CTEPH PH patients, such as those with idiopathic PAH or PH associated with interstitial lung disease, diffuse parenchymal remodeling may also reduce Zeff through mechanisms such as increased low-Z fibrotic tissue or air trapping. As a result, despite differing perfusion patterns, both groups can exhibit similar overall Zeff values, which helps explain why Zeff was significantly lower in the PH patients compared to the healthy controls but not significantly different between the CTEPH and non-CTEPH groups.

Therefore, while the iodine content influences Zeff, the absence of a significant difference in Zeff between CTEPH and non-CTEPH may reflect the interplay of multiple factors, including perfusion heterogeneity, systemic collaterals, and parenchymal composition. These findings further emphasize that iodine density and Zeff are non-redundant but complementary parameters, with each offering distinct physiologic and structural insights into pulmonary vascular disease.

## 5. Limitations

Despite these promising results, this study had several limitations. The sample size, particularly in the CTEPH subgroup, was relatively small, which might affect the generalizability of the findings. Additionally, the manual placement of ROIs introduces potential variability that could influence the reproducibility of the measurements across different operators or institutions. Furthermore, due to the limited sample size, achieving balanced distributions of gender and age was challenging. These demographic factors could act as confounding variables, potentially influencing this study’s outcomes. Although the Mann–Whitney U test was employed to compare groups, this non-parametric test does not account for such covariates, which may limit the robustness of the conclusions. Future studies could address these limitations by employing automated ROI technologies and recruiting larger, more diverse cohorts to enable the use of statistical methods that adjust for potential confounding factors.

## 6. Conclusions

Our study demonstrated the parametric characteristics of pulmonary hypertension on spectrum CT. It may aid in identifying PH cases from the normal population as well as CTEPH cases among PH patients. By leveraging advanced imaging metrics like iodine density and effective atomic numbers, clinicians may achieve a more integrated and comprehensive assessment of PH and its possible subgroups, providing a more targeted and optimized treatment strategy in the future.

## Figures and Tables

**Figure 1 diagnostics-15-01069-f001:**
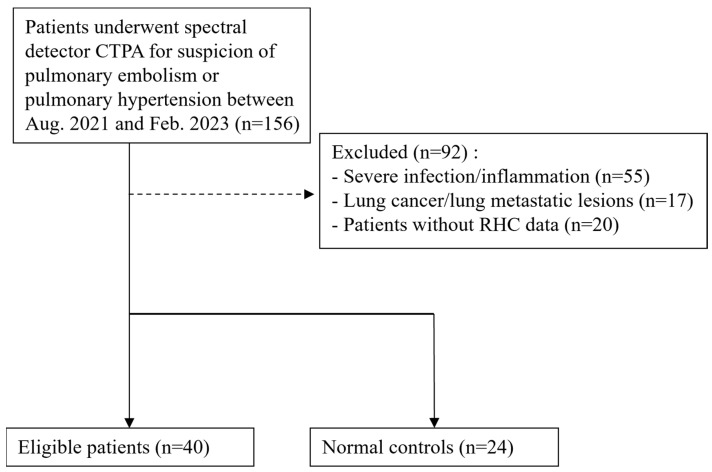
Flowchart of enrollment of PH patients and normal controls.

**Figure 2 diagnostics-15-01069-f002:**
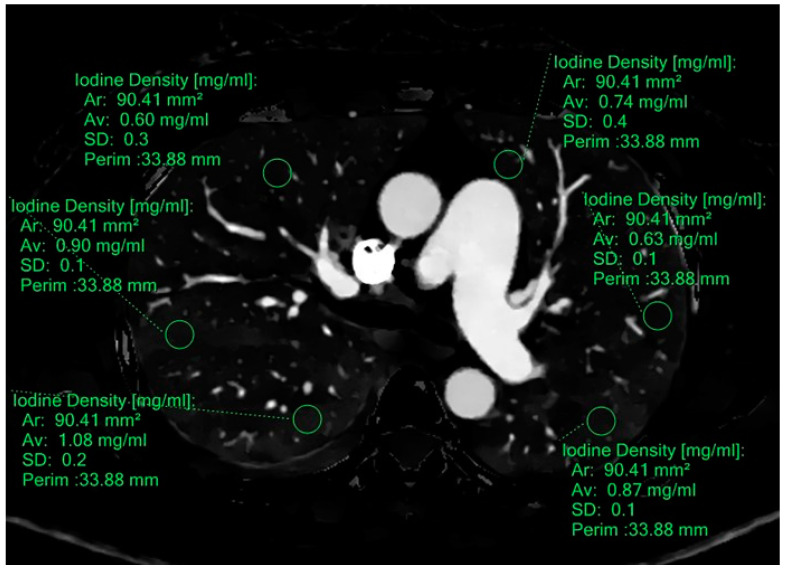
Iodine density (mean and standard deviation) measured by manually placing an 80~100 mm^2^ standardized ROI over 24 set areas on the axial iodine density map images, avoiding major vessels; these were placed at the anterior, mid, and posterior regions on axial images at 4 predefined levels (1, the aortic arch; 2, the carina; 3, the right pulmonary artery; 4, halfway between level “3” and the diaphragm). This image displays the level at the carina.

**Figure 3 diagnostics-15-01069-f003:**
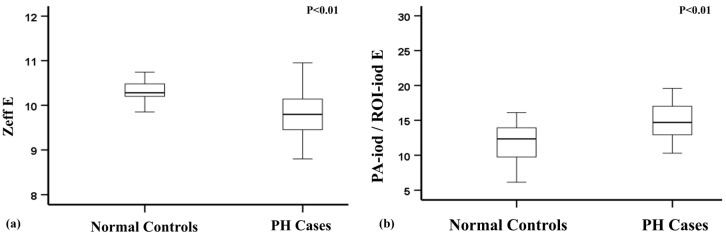
Boxplots illustrating the distribution of the effective atomic number (Zeff) and PA-iod/ROI-iod of the total lung average (Level E) in the PH and non-PH control patients. (**a**). The effective atomic number was significantly lower in the PH group than in the normal controls. (**b**). A significantly higher PA-iod/ROI-iod ratio was observed in the PH group compared to the normal controls.

**Figure 4 diagnostics-15-01069-f004:**
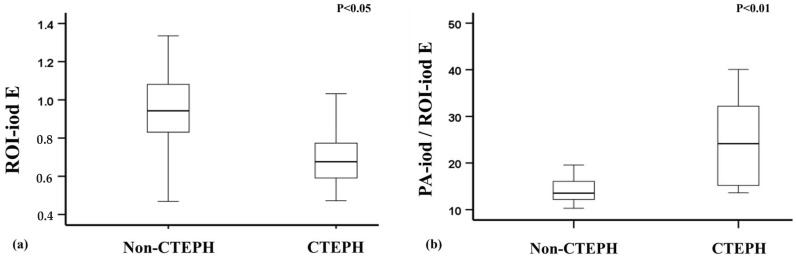
Boxplots illustrating the distribution of the iodine density and PA-iod/ROI-iod of the total lung average (E) in the CTEPH and non-CTEPH patients. (**a**). Iodine density was significantly higher in the non-CTEPH group than in the CTEPH group. (**b**). A significantly lower PA-iod/ROI-iod ratio was observed in the non-CTEPH group compared to the CTEPH group.

**Figure 5 diagnostics-15-01069-f005:**
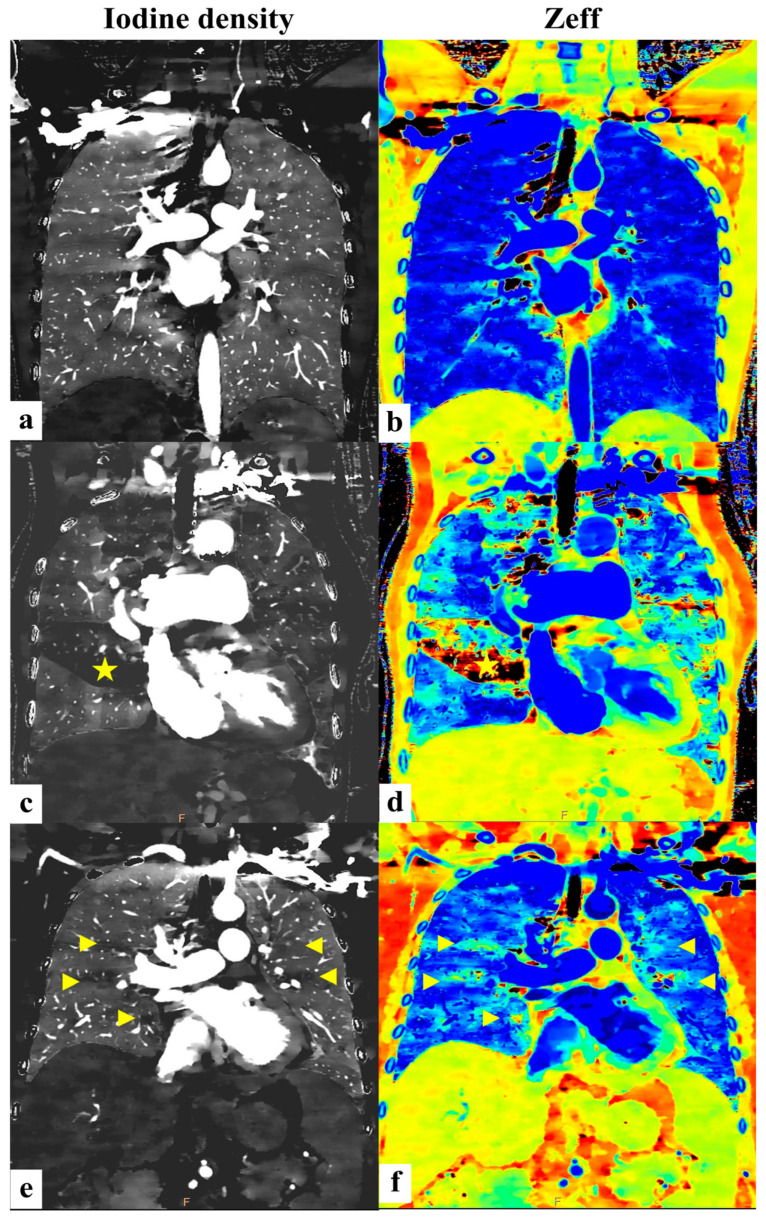
(**a**,**b**) (normal controls): Bilateral lung parenchyma with uniform iodine density/Zeff demonstrating a homogeneous image pattern. (**c**,**d**) (CTEPH group): Patch region of lung parenchyma with decreased iodine density/Zeff (yellow asterisk) indicating an area with suspicious decreased pulmonary perfusion. (**e**,**f**) (non-CTEPH pulmonary hypertension group): Scattered small areas of diminished iodine density/Zeff (yellow arrowheads) interspersed among normal lung parenchyma, revealing a very heterogeneous pattern.

**Table 1 diagnostics-15-01069-t001:** Demographics of the study groups.

	PH Case (n = 33)	Control (n = 24)
Age	62 [44.0–80.5]	60 [39.3–71.5]
Sex		
Female	23 [69.7%]	12 [50%]
Male	10 [30.3%]	12 [50%]

**Table 2 diagnostics-15-01069-t002:** PH patient subgroup (N = 33).

	Group 1	Group 2	Group 3	Group 4
Num. of patients	8 (24.2%)	7 (21.2%)	9 (27.3%)	9 (27.3%)

**Table 3 diagnostics-15-01069-t003:** SDCT parameters of the PH group and the control group.

	Normal Control (n = 24)	PH Case (n = 33)	*p*-Value
Sex			0.132
Female	12 (50.0%)	23 (69.7%)	
Male	12 (50.0%)	10 (30.3%)	
ROI-iod A	1.03 (0.88–1.19)	0.94 (0.76–1.21)	0.207
ROI-iod B	1.02 (0.79–1.10)	0.91 (0.67–1.03)	0.091
ROI-iod C	0.99 (0.83–1.16)	0.81 (0.68–1.07)	0.037 *
ROI-iod D	0.87 (0.72–1.01)	0.83 (0.66–1.04)	0.884
ROI-iod E	0.98 (0.82–1.13)	0.91 (0.69–1.06)	0.139
Zeff A	10.46 (10.37–10.82)	9.94 (9.68–10.30)	<0.001 **
Zeff B	10.30 (10.07–10.55)	9.95 (9.59–10.26)	0.003 **
Zeff C	10.38 (10.15–10.60)	9.83 (9.42–10.14)	<0.001 **
Zeff D	10.14 (9.92–10.51)	9.87 (9.14–10.06)	0.001 **
Zeff E	10.28 (10.20–10.49)	9.80 (9.44–10.15)	<0.001 **
PA-iod/ROI-iod A	11.92 (8.48–13.34)	13.55 (11.18–16.96)	0.011 *
PA-iod/ROI-iod B	12.20 (10.80–14.73)	15.01 (13.39–19.85)	0.003 **
PA-iod/ROI-iod C	11.89 (9.54–13.82)	14.22 (12.59–21.58)	0.001 **
PA-iod/ROI-iod D	12.69 (10.29–17.39)	15.57 (12.78–19.75)	0.047 *
PA-iod/ROI-iod E	12.34 (9.64–14.04)	14.70 (12.61–17.49)	0.005 **
Aa-iod/ROI-iod A	9.82 (7.34–12.37)	10.13 (6.42–12.96)	0.808
Aa-iod/ROI-iod B	10.52 (8.52–12.88)	11.45 (7.49–13.02)	0.923
Aa-iod/ROI-iod C	9.99 (7.88–12.47)	10.47 (7.57–13.20)	0.722
Aa-iod/ROI-iod D	10.82 (9.61–13.95)	10.99 (7.74–12.53)	0.316
Aa-iod/ROI-iod E	10.09 (8.14–12.53)	10.88 (7.26–12.95)	0.987

Chi-square test or Mann–Whitney U test, median (IQR). * *p* < 0.05, ** *p* < 0.01. ROI: region of interest; iod: iodine density; Zeff: effective Z; PA: pulmonary trunk; Aa: ascending aorta

**Table 4 diagnostics-15-01069-t004:** SDCT parameters of the CTEPH and non-CTEPH groups.

	Non-CTEPH (n = 24)	CTEPH (n = 9)	*p*-Value
Sex			1.000
Female	17 (70.8%)	6 (66.7%)	
Male	7 (29.2%)	3 (33.3%)	
ROI-iod A	1.04 (0.83–1.26)	0.76 (0.67–0.87)	0.006 **
ROI-iod B	0.96 (0.73–1.10)	0.72 (0.63–0.92)	0.043 *
ROI-iod C	0.84 (0.74–1.07)	0.63 (0.49–0.85)	0.035 *
ROI-iod D	0.95 (0.75–1.13)	0.65 (0.46–0.86)	0.026 *
ROI-iod E	0.94 (0.82–1.09)	0.68 (0.58–0.87)	0.015 *
Zeff A	9.94 (9.66–10.27)	10.03 (9.66–10.45)	0.872
Zeff B	9.92 (9.52–10.29)	10.00 (9.63–10.40)	0.657
Zeff C	9.69 (9.40–10.04)	9.96 (9.52–10.36)	0.210
Zeff D	9.79 (9.35–10.00)	9.87 (8.90–10.22)	0.903
Zeff E	9.79 (9.43–10.12)	9.94 (9.46–10.30)	0.686
PA-iod/ROI-iod A	13.16 (10.90–14.64)	20.32 (14.0131.74)	0.005 **
PA-iod/ROI-iod B	13.83 (12.33–16.74)	21.33 (15.88–29.70)	0.005 **
PA-iod/ROI-iod C	13.43 (12.26–17.59)	24.25 (15.23–42.08)	0.012 *
PA-iod/ROI-iod D	13.81 (12.05–17.66)	27.52 (15.13–40.81)	0.010 *
PA-iod/ROI-iod E	13.54 (12.17–16.21)	24.16 (14.56–33.63)	0.006 **
Aa-iod/ROI-iod A	9.41 (6.18–12.65)	11.31 (9.33–13.79)	0.196
Aa-iod/ROI-iod B	9.45 (7.48–13.48)	12.00 (9.30–13.01)	0.544
Aa-iod/ROI-iod C	9.82 (6.87–12.78)	13.14 (9.42–16.33)	0.124
Aa-iod/ROI-iod D	9.59 (7.11–12.23)	11.88 (9.65–6.29)	0.124
Aa-iod/ROI-iod E	9.62 (6.55–12.56)	12.71 (9.57–13.15)	0.196

Chi-square test or Mann–Whitney U test, median (IQR). * *p* < 0.05, ** *p* < 0.01. ROI: region of interest; iod: iodine density; Zeff: effective Z; PA: pulmonary trunk; Aa: ascending aorta.

## Data Availability

The raw data supporting the conclusions of this article will be made available by the authors on request.

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
