# Peer review of "Investigating Patients with Pulmonary Hypertension Under Detector-Based Spectral Computed Tomography"

_diagnostics, 2025, doi:10.3390/diagnostics15091069_

Round 1
Reviewer 1 Report
Comments and Suggestions for Authors
The manuscript presents a valuable study evaluating the effectiveness of spectral detector computed tomography (SDCT) in assessing pulmonary hypertension (PH) and chronic thromboembolic pulmonary hypertension (CTEPH). The authors effectively investigate iodine density and effective atomic number (Zeff) in lung parenchyma, comparing these parameters between PH patients and healthy controls, as well as between CTEPH and non-CTEPH patients.
The study successfully demonstrates significant differences in iodine density and Zeff between the PH and control groups, as well as between CTEPH and non-CTEPH patients. However, several limitations merit attention:
- Sample size constraints potentially limit statistical power and generalizability
- ROI measurement variability raises questions about reproducibility
- Potential confounding factors and biases may affect result interpretation
- Clinical implications require further elaboration
Specific Recommendations for Revision:
- Discussion Revision:
- The discussion section should be restructured to directly relate to the presented results. The authors should focus on contextualizing their findings within existing literature and highlighting their clinical significance.
- Please use general types of dual energy CT models (such as dual-source, detector-based, rapid kVp switching) rather than specific CT machine names.
- Figures Inclusion: The manuscript would benefit significantly from including figures that clearly illustrate the differences in pulmonary parameters between CTEPH patients and normal controls. Visual representations of the quantitative differences in iodine density and Zeff would enhance reader comprehension and better demonstrate the clinical relevance of these findings.
end of comments
Author Response
Reviewer 1:
1. “The discussion section should be restructured to directly relate to the presented results. The authors should focus on contextualizing their findings within existing literature and highlighting their clinical significance.”
Response:
Thank you for this valuable feedback. In response, we have revised the Discussion section to more closely align with the study results and better integrate them into the existing body of literature. Specifically:
-
Contextualization with Recent Studies: We incorporated comparisons with recent DECT studies on pulmonary hypertension, highlighting how our findings regarding iodine density ratios and effective atomic number (Zeff) align with or diverge from prior results. This strengthens the relevance and validity of our observations.
-
Emphasis on Clinical Relevance: We expanded on the potential clinical utility of DECT-derived parameters such as PA-iod/ROI-iod and Zeff, proposing their role as non-invasive imaging biomarkers for assessing perfusion abnormalities and aiding in subtype differentiation, particularly in distinguishing CTEPH from other forms of PH.
These revisions aim to clarify the significance of our results and situate them meaningfully within the current scientific landscape.
2. “The manuscript would benefit significantly from including figures that clearly illustrate the differences in pulmonary parameters between CTEPH patients and normal controls.”
Response:
We appreciate the reviewer’s suggestion. In response, we have added a new set of figures (Figure 5, page 8) to visually demonstrate the differences in iodine density and Zeff between CTEPH patients and normal controls. These comparative images are intended to improve reader comprehension of the observed perfusion patterns and enhance the visual presentation of our key findings.
Reviewer 2 Report
Comments and Suggestions for Authors
PH is a extremely complex disease, leading to different cause. Actually, according to guidelines, the only instrument to diagnosis PH is RHC, not always recommended.
The introduction of novel instrument, especially if non-invasive, for diagnosis and follow up of PH is necessary in clinical practice.
That’s why I think that this original article is very interesting and a promising starting point.
Unfortunately, the sample is very small, but we have also to take in account that PH, and in particular pulmonary arterial hypertension and CTEPH are low prevalence disease, often difficult to diagnose.
Here my comment:
Introduction
Line 37-39: Please specify that PH is diagnosed only via left heart catheterization
Can lung parenchymal distortion influence the interpretation of DECT? If so, are there any patient with particular lung parenchymal abnormalities in the two sub-group? I see that patients with >50% emphysema were excluded, what about patients with interstitial lung disease?
Materials and Methods
2.1 patient population
Line 102-106: as mentioned before, mPAP cut-off for diagnosis of PH is 20 mmHg, could you explain why you choose patient with mPAP> 25 mmHg?
2.3. Statistical Analysis
As the two groups are not matched for gender and age, the statistical model may be deficient, have measures been taken to address this, e.g. by applying logistic regression models?
Results
In Table 2. Please add also % for any sub-group
In the sub-group of PH patients, which and how many patients were under treatment, where applicable according to guidelines?
In the group of CTEPH patients, did someone already received surgical treatment?
thank you
Author Response
Reviewer 2:
1. “Line 102–106: As mentioned before, mPAP cut-off for diagnosis of PH is 20 mmHg. Could you explain why you chose patients with mPAP > 25 mmHg?”
Response:
We appreciate the reviewer’s question regarding our choice of the mPAP threshold. Although the 6th World Symposium on Pulmonary Hypertension has recommended lowering the diagnostic cut-off to >20 mmHg based on updated hemodynamic insights, our institution’s cardiology team continues to apply the traditional threshold of ≥25 mmHg when performing right heart catheterization. This approach prioritizes diagnostic specificity and consistency with earlier classification criteria, helping to avoid overdiagnosis in borderline cases. Accordingly, we adopted the ≥25 mmHg cut-off in this study to reflect actual clinical practice at our center and to ensure comparability with prior studies based on the same definition.
2. “Can lung parenchymal distortion influence the interpretation of DECT? If so, are there any patients with particular lung parenchymal abnormalities in the two subgroups?”
Response:
Yes, we recognize that parenchymal distortion—particularly from severe emphysema—can interfere with accurate ROI placement on DECT images. Patients with extensive emphysematous changes that prevented ROI placement at all four predefined levels were excluded. In cases with interstitial lung disease (ILD), including patterns such as UIP or NSIP, we avoided selecting ROIs from areas with marked fibrosis or architectural distortion. Our rationale was that PH tends to affect the lungs globally, and excluding severely affected regions improves the reliability of overall perfusion assessment while reducing local bias introduced by distortion.
3. “As the two groups are not matched for gender and age, the statistical model may be deficient. Have measures been taken to address this, e.g., by applying logistic regression models?”
Response:
We thank the reviewer for raising this important concern. Due to the modest sample size, exact matching on demographic variables such as age and gender was not feasible. We used the Mann–Whitney U test for intergroup comparisons, as it is a robust non-parametric method appropriate for small and non-normally distributed datasets. While we acknowledge that multivariate methods like logistic regression would allow adjustment for potential confounding variables, applying them in our study risked model overfitting. We have now acknowledged this limitation in the revised manuscript and recommend that future research with larger sample sizes consider such methods to improve statistical rigor.
4. “In the subgroup of PH patients, which and how many patients were under treatment, where applicable according to guidelines? In the group of CTEPH patients, did someone already receive surgical treatment?”
Response:
All patients in our study were newly diagnosed with PH at the time of spectral CT imaging. None of the non-CTEPH patients had initiated targeted pharmacologic therapy before undergoing DECT. In the CTEPH group, all patients had previously received anticoagulant treatment as part of their management for pulmonary embolism, but none had undergone pulmonary endarterectomy (PEA) or balloon pulmonary angioplasty (BPA) prior to imaging. We have clarified this information in the revised manuscript (Section 3.1, Study Population).